

**Enhanced heterogeneous uptake of sulfur dioxide on**
**mineral particles through modification of iron speciation**
**during simulated cloud processing**
**Zhenzhen Wang[1], Tao Wang[1], Hongbo Fu[1, 2, 3], Liwu Zhang[1], Mingjin Tang[4],**
**Christian George[5], Vicki H. Grassian[6], and Jianmin Chen[1]**
[1]Shanghai Key Laboratory of Atmospheric Particle Pollution and Prevention,
Department of Environmental Science & Engineering. Institute of Atmospheric
Sciences, Fudan University, Shanghai, 200433, China
[2]Shanghai Institute of Pollution Control and Ecological Security, Shanghai 200092,
China
[3]Collaborative Innovation Center of Atmospheric Environment and Equipment
Technology (CICAEET), Nanjing University of Information Science and Technology,
Nanjing 210044, China
[4]State Key Laboratory of Organic Geochemistry and Guangdong Key Laboratory of
Environmental Protection and Resources Utilization, Guangzhou Institute of
Geochemistry, Chinese Academy of Sciences, Guangzhou 510640, China
[5]University of Lyon, Université Claude Bernard Lyon 1, CNRS, IRCELYON,
F-69626, Villeurbanne, France
[6]Departments of Chemistry and Biochemistry, University of California, San Diego, La
Jolla, California 92093, United States
*Correspondence to*: Hongbo Fu (fuhb@fudan.edu.cn); Jianmin Chen
(jmchen@fudan.edu.cn)



**Abstract.** Iron-containing mineral aerosols play a key role in the oxidation of sulfur
species in the atmosphere. Simulated cloud processing (CP) of typical mineral
particles, such as illite (IMt-2), nontronite (NAu-2), smectite (SWy-2) and Arizona
test dust (ATD) is shown here to modify sulfur dioxide ($SO_2$) uptake onto mineral
surfaces. Heterogeneous oxidation of $SO_2$ on particle surfaces was firstly investigated
using an *in situ* DRIFTS apparatus. Our results showed that the BET surface area
normalized uptake coefficients ($\gamma_{BET}$) of $SO_2$ on the IMt-2, NAu-2, SWy-2 and ATD
samples after CP were 2.2, 4.1, 1.5 and 1.4 times higher than the corresponding ones
before CP, respectively. The DRIFTS results suggested that CP increased the amounts
of reactive sites (e.g., surface OH groups) on the particle surfaces and thus enhanced
the uptake of $SO_2$. TEM showed that the particles broke up into smaller pieces after
CP, and thus produced more active sites. The "free-Fe" measurements confirmed that
more reactive Fe species were present after CP, which could enhance the $SO_2$ uptake
more effectively. Mössbauer spectroscopy further revealed that the formed Fe phase
were amorphous Fe(III) and nanosized ferrihydrite hybridized with Al/Si, which were
possibly transformed from the Fe in the aluminosilicate lattice. The modification of
Fe speciation was driven by the pH-dependent fluctuation coupling with Fe
dissolution-precipitation repeatedly during the experiment. Considering both the
enhanced $SO_2$ uptake and subsequent promotion of iron dissolution along with more
active Fe formation, which in turn lead to more $SO_2$ uptake, it was proposed that there
may be a positive feedback between $SO_2$ uptake and iron mobilized on particle
surfaces during CP, thereby affecting climate and biogeochemical cycles. This
self-amplifying mechanism generated on the particle surfaces may also serve as the
basis of high sulfate loading in severe fog-haze events observed recently in China.



## 1 Introduction

Mineral dust is a major fraction of global atmospheric aerosol budget, with an
estimated annual emission flux of 1000 to 3000 Tg into the atmosphere (Jickells et al.,
2005; Andreae and Rosenfeld, 2008). Mineral dust aerosol mainly consists of quartz,
feldspars, carbonates (calcite, dolomite), and clay minerals (illite, kaolinite, chlorite,
montmorillonite), the exact composition varies with source (Claquin et al., 1999;
Formenti et al., 2008). During the long-range transport, mineral dust provides a
reactive surface for heterogeneous chemistry (Zhang et al., 2006; George et al., 2015;
Huang et al., 2015). Heterogeneous reactions of atmospheric trace gases on mineral
dust particles are of great significance as these reactions alter the chemical balance of
the atmosphere and modify the properties of individual particles (Usher et al., 2003;
Wu et al., 2011; Huang et al., 2015).
$SO_2$ is an important trace gas, which is released mainly by fossil fuel combustion
and volcanic emission. The heterogeneous conversion of $SO_2$ on mineral dust surfaces
leads to the formation of sulfuric acid and sulfate aerosols, resulting in a significant
cooling effect on the global climate by scattering solar radiation and acting as cloud
condensation nuclei (CCN) to affect climate indirectly (Lelieveld and Heintzenberg,
1992; Usher et al., 2003; Kolb et al., 2010). In addition, sulfate containing particles
play a significant role in the haze formation in China in recent years (Sun et al., 2014;
Wang et al., 2014; Yang et al., 2017). $SO_2$ can be gaseous oxidized to sulfate by OH
radical, and be aqueous oxidation in cloud and fog droplets by ozone and hydrogen
peroxide (Luria and Sievering, 1991), or through heterogeneous processes that occur
on aerosol particle surfaces (Usher et al., 2003; Ullerstam et al., 2003). However, the
high sulfate levels measured in recent field observations cannot be explained by
current atmospheric models (Kerminen et al., 2000; Wang et al., 2003; Cheng et al.,


2016), leading to a large gap between the modeled and field-observed sulfate
concentrations using known oxidation pathways (Herman, 1991; Kasibhatla et al.,
1997; Barrie et al., 2016). Overall, on a global scale, atmospheric $SO_2$ concentration
were typically overestimated, while sulfate tended to be underestimated, suggesting
missing sulfate production pathways (Harris et al., 2013; Kong et al., 2014).
It has been suggested that the heterogeneous conversion of $SO_2$ could make an
important contribution to the atmospheric sulfate loading. Laboratory studies typically
focus on $SO_2$ uptake onto a variety of metal oxides and mineral particles (Goodman et
al., 2001; Usher et al., 2002; Zhao et al., 2015; Yang et al., 2016), and have confirmed
that its conversion rate on the surface of Fe (hydr)oxides was faster compared to other
metal oxides investigated, in good agreement with the field-measurement (Usher et al.,
2002; Zhang et al., 2006). Atmospheric Fe is emitted from both anthropogenic
(primarily biomass burning, coal and oil combustion) and natural (mineral dust and
volcanic ash) sources, with the mineral dust source dominant globally (Siefert et al.,
1998; Luo et al., 2008). It has been established that an important in cloud S (IV)
oxidation pathway is catalyzed by natural transition metal ions, especially Fe hosted
within mineral particles (Alexander et al, 2009; Harris et al., 2013; Ito et al., 2019).
Another important consideration for heterogeneous chemistry of mineral dust
aerosol, is how mineral dust particles change in the atmosphere. During long-range
transport, mineral particles often undergo chemical ageing by atmospheric processes
(Mahowald et al., 2005; Baker and Croot, 2010; Shi et al., 2011). Cloud processing
involves cloud water condensation and evaporation on the particle surfaces, along
with drastic liquid water content and pH fluctuations (Mackie, 2005; Shi et al., 2011;
Rubasinghege et al., 2016). During CP, the high relative humidity (RH) results in high
aerosol water content and relatively high pH (Behra et al., 1989; Baker and Croot,



2010; Shi et al., 2011). While water evaporation from cloud droplets to wet aerosol at
higher temperature, the particles only contain a concentrated aqueous aerosol solution,
in which the pH can be lower than 2 (Zhu et al., 1993; Meskhidze, 2003; Shi et al.,
2015). Therefore, there is a highly acidic film (e.g., pH = 2) in the "wet aerosol" phase
versus a less acidic droplet (near-neutral, 5-6) in the "cloud droplet" phase within
clouds (Shi et al., 2015). During its lifetime, a typical aerosol particle may experience
several cloud cycles involving large pH variations before being removed from the
atmosphere as rain or through dry deposition (Pruppacher and Jaenicke, 1995; Maters
et al., 2016). Herein, the simulated CP experiment was conducted by changing pH
between 2 and 5-6, in accordance with the previous studies (Spokes et al., 1994;
Mackie, 2005; Shi et al., 2009).
It was well documented that pH is especially important for Fe mobilization (Zhu et
al., 1993; Desboeufs et al., 2001; Deguillaume et al., 2010; Maters et al., 2016). The
fluctuating pH during CP will impact and change the Fe speciation and morphology in
dust particles (Zhuang et al., 1992; Wurzler et al., 2000; Shi et al., 2009; Kadar et al.,
2014). The low pH will increase Fe solubility and bioavailability of dust during
transport, thereby providing Fe external input to the open ocean surface to promote
marine prime productivity (Spokes et al., 1994; Desboeufs et al., 2001). It has been
found that Fe-rich nanoparticle aggregates were formed from Saharan soil and
goethite upon simulated CP conditions, in good agreement with their
field-measurements from the wet-deposited Saharan dusts collected from the western
Mediterranean (Shi et al., 2009). Fe nanoparticles are more chemically reactive
(Wurzler et al., 2000; Desboeufs et al., 2001), possibly lead to a remarkable difference
in heterogeneous chemistry. However, little is known about the influence of CP on
$SO_2$ uptake onto particle surfaces up to now.


In this study, we employed four typical mineral samples as surrogates to perform
simulated CP experiments. The $SO_2$ uptakes on the mineral particles before and after
CP were compared using *in situ* diffuse reflectance infrared Fourier transform
spectroscopy (DRIFTS). Transmission electron microscopy (TEM) was applied to
observe the morphological and mineralogical change of mineral particles. The Fe
speciation modification during simulated CP was further monitored by the dissolved
Fe    measurement,    the    "free-Fe"    analysis    and    Mössbauer    spectroscopic
characterization.
**2    Material and methods**
**2.1 Mineral particles**
The standard mineral samples of IMt-2, NAu-2 and SWy-2 were purchased from
the Source Clay Minerals Repository (Purdue University, West Lafayette, IN). ATD
was purchased from Powder Technology Inc. (Burnsville, MN, USA). The mineral
samples were coarsely ground using a mortar and pestle before being more finely
ground using an All-dimensional Planetary Ball Mill QM-QX (Nanjing University
Instrument Plant) and were sieved to particle diameters ($D_p$) < 45 $\mu$m prior to analysis.
The Brunauer-Emmett-Teller specific surface areas ($S_{BET}$) of the samples were
measured with a Quantachrome Nova 1200 BET apparatus. Total iron content ($Fe_T$) of
the samples were determined using an inductively coupled plasma atomic emission
spectroscopy (ICP-AES, Jobin Yvon Ultima). The chemical compositions of the
particles were analyzed by X-ray fluorescence spectrometry (XRF, PANalytical Axios
Advanced).
**2.2 Cloud processing simulation experiment**
The simulated CP experiments were conducted at a constant temperature ($298 \pm 1K$)
using a Pyrex glass vessel with a water jacket. The suspensions contained a mineral


particle loading of 1 g $L^{-1}$ were subjected to acidic (pH = 2 ± 0.1, 24 h) and
near-neutral pH (pH = 5-6, 24 h) cycles for 1-3 times according to the previous
methods (Spokes et al., 1994; Mackie, 2005; Shi et al., 2009). The suspension pH was
adjusted by adding dilute $H_2SO_4$ or $NH_4OH$. The $CaCO_3$ equivalent alkalinity of the
dust was determined in accordance with APHA method 2320B so that acid additions
to control pH could be adjusted accordingly (Mackie, 2005). The amount of acid or
alkali added to achieve these pH cycles was less than 1% of the total volume of the
suspensions. The experiments were performed under a constant stirring (about 50
rpm) in the dark for 144 h. At the end of the CP experiment, the suspensions were
filtered through 0.2 $\mu$m PTFE filters (Millipore). The filter residue was air-dried, and
was further applied to the DRIFTS experiment, as well as TEM observation, "free-Fe"
measurement and Mössbauer spectroscopic characterization.
**2.3 $SO_2$ uptake on the mineral particles**
The $SO_2$ uptake on the particle surfaces before and after CP was investigated by a
Shimadzu Tracer-100 FTIR spectrometer equipped with a high-sensitivity mercury
cadmium telluride (MCT) detector and a diffuse reflectance accessory. A temperature
controller was fitted to the DRIFTS chamber to ensure constant reaction temperature
(298 K). Weighted sample was placed into a ceramic crucible (0.35 mm depth, 5 mm i.
d.) in the chamber. Mass flow controllers (Beijing Sevenstar electronics Co., LTD)
were used to adjust the reactant gases to a flux with expected concentration and
relative humidity. The sample was firstly pretreated in a 100 mL $min^{-1}$ flow of
synthetic air (21% $O_2$ and 79% $N_2$) for 1 h to blow off water and impurities on
particle surface. When the background spectrum of the fresh sample reached steady
state, the reactant gas of $SO_2$ (5.0 ppm) along with synthetic air was introduced into
the chamber at a total flow rate of 120 mL $min^{-1}$ for 45 min, during which the IR



spectrum was recorded automatically every 5 min at a resolution of 4 cm$^{-1}$ for 100
scans in the spectral range of 900 to 4000 cm$^{-1}$. Atmospheric moisture was simulated
with a RH level around 40 % by guiding one high-pure air flux through water. The
humidity value was monitored using a hygrometer.
The sulfate products were analyzed by ion chromatography (IC) after the DRIFTS
experiments. The particles were extracted with 5 ml ultrapure water by ultrasonic
extractor. After 10 min, the extracted solution was passed through a 0.22 $\mu$m PTFE
membrane filter and the leaching solution was analyzed using a Metrohm 883 Basic
IC equipped with an A5-250 column. A weak base eluent (3.2 mmol L$^{-1}$ Na$_2$CO$_3$ plus
1.0 mmol L$^{-1}$ NaHCO$_3$) was used for anion detection at a flow rate of 0.70 ml min$^{-1}$.
To discriminate the adsorbed sulfate during simulated CP experiment and the sulfate
ions generated from the heterogeneous reaction, the adsorbed sulfate on the particles
during simulated CP experiment were initially measured as blank. The heterogeneous
uptake of SO$_2$ was calculated by subtracting the blank value from the total sulfate
ions.
The reactive uptake coefficient ($\gamma$) was defined as the rate of sulfate formation on
the surface (d[SO$_4^{2-}$]/dt, ions s$^{-1}$) divided by collision frequency ($Z$, molecules s$^{-1}$)
(Usher et al., 2003; Ullerstam et al., 2003; Kong et al., 2014; Huang et al., 2015).
$$\gamma = \frac{d[SO_4^{2-}]/dt}{Z}, \qquad (1)$$
$$Z = \frac{1}{4} \times A_s \times [SO_2] \times v, \qquad (2)$$
$$v = \sqrt{\frac{8RT}{\pi M_{SO_2}}}, \qquad (3)$$
Where, $A_s$ is the effective sample surface of the samples, m$^2$; $v$ is the mean
molecular velocity of SO$_2$, m s$^{-1}$; $R$ is the gas constant, J mol K$^{-1}$; $T$ is the absolute
temperature, K; and $M_{SO_2}$ is the molecular weight of SO$_2$, kg mol$^{-1}$.



A conversion factor was obtained by a calibration plot with numbers of $SO_4^{2-}$
analyzed by ion chromatography (IC, Metrohm 883 Basic, Switzerland) versus the
integrated areas of sulfate products from DRIFTS spectra. The residual sulfate during
simulated CP experiments were deducted as background. The calculated conversion
factor of $SO_4^{2-}$ is $1.170 \times 10^{15}$ (ions · integrated units$^{-1}$). Integrated areas for the total
sulfur-containing products were calculated to show the maximal sulfate formation
rates. The reactive uptake coefficient for $SO_2$ was determined to be $\gamma_{BET}$ and $\gamma_{geo}$ using
the BET area ($A_{BET}$ = mass×$S_{BET}$) and geometric area ($A_{geo}$ = mass×$S_{BET}$) as the
reactive area, respectively.
**2.4 Morphological and mineralogical characterization of the Fe speciation**
A FEI TECNAI G2 S-TWIN F20 TEM equipped with an Oxford energy-dispersive
X-ray spectrometer (EDX) was used to analyze the morphological and chemical
composition of individual particles before and after CP. Suspensions (0.2 g L$^{-1}$) of
each particle were prepared in methanol and sonicated for at least 1 h. A drop of this
suspension was then applied to a carbon-coated Cu TEM grid (400 mesh; EMS). A
FEI TECNAI G2 S-TWIN F20 TEM equipped with an Oxford energy-dispersive
X-ray spectrometer (EDX) was used for high-resolution imaging and to analyze the
chemical composition of individual particles. The Fe content of the typical individual
mineral particle were calculated from the values of 50 typical particles. Selected area
electron diffraction (SAED) was used to identify the crystalline phases.
The content of "free-Fe" in the mineral particles was determined by a
citrate-buffered-dithionite (CBD) sequential Fe extractions method according to the
literature (Lafon et al., 2004; Shi et al., 2009). Simply, 30 mg of the dust samples
were treated for 24 h with a 10 mL ascorbate solution (pH = 7.5) to extract chemically
highly labile Fe phases (Fe$_A$), mainly composed of amorphous, nanoparticle and/or





poorly crystalline ferrihydrite. The solutions were filtered through 0.2 $\mu$m
polycarbonate filters. The dust particles collected on the filters were subsequently
treated for 2 h with a 10 mL sodium dithionite solution (pH = 4.8) to extract
crystalline Fe (oxyhydr) oxides (Fe$_D$), which are mainly goethite and hematite. After
each reaction step, the dissolved Fe concentrations (Fe$_A$ and Fe$_D$) in the filtrates were
determined using ICP-AES. The sum of these two pools (Fe$_A$ + Fe$_D$) was defined as
the "free-Fe" fraction (Shi et al., 2011). The other fraction was donated as the
"structural-Fe" in aluminosilicate crystals, which could be calculated from the
difference between the Fe$_T$ and "free-Fe" fractions (Lafon et al., 2004).

The Mössbauer spectroscopic analysis performed in transmission geometry with a

constant acceleration was used to inspect the chemical valence and the surrounding
structure of Fe in the particles before and after CP. $^{57}$Co was used as the Mössbauer
source, and a 1 mm thick Na(TI) scintillator coupled to a EMI9750B photoelectric
multiplier was used as the detector (Cwiertny et al., 2008). The measurement was
carried out at room temperature (RT) with a duration of 24 hours for one sample
(around $1.5 \times 10^6$ counts per channel). Experimental data were fitted by a least-squares
fitting-program. The isomer shift values were calibrated against a spectrum for α-Fe
metal foil.

During the simulated CP experiment, the total dissolved iron (Fe$_s$) and the dissolved

Fe(II) in the suspensions were measured colorimetrically by the Ferrozine method, as
described in previous studies (Viollier et al., 2000; Cwiertny et al., 2008). For Fe(II)
analysis, 200 mL of a 5 mM 1, 10-phenanthroline solution and 200 mL of an
ammonium acetate buffer were added into 1 mL of sample. To avoid possible
interference from Fe(III), which can also form a complex with 1,10-phenanthroline
when present at high concentrations, 50 mL of 0.43 M ammonium fluoride was added


to the sample prior to 1,10-phenanthroline. The mixture was allowed to sit in the dark
for 30 min prior to ultraviolet-visible spectroscopy (UV-Vis) analysis, during which
time a reddish-orange color developed if Fe(II) was present. $Fe_s$ was determined via
the same protocol, except that 20 mL of 1.5 M hydroquinone, which reduces Fe(III) to
Fe(II), was added to the sample rather than ammonium fluoride. Absorbance
measured at 510 nm was converted to concentrations using aqueous standards
prepared from anhydrous beads of ferrous chloride. Standards were prepared in each
acid used in dissolution studies, and no matrix effects were observed. These
conditions resulted in a detection limit of 1 $\mu$M. The concentration of dissolved
Fe(III) was calculated from the difference in experimentally measured concentrations
of total dissolved iron and dissolved Fe(II).
Additionally, the dissolved Fe(III) could precipitate out as the pH increased, and
then the Fe mineraology of the deposit was also observed. NAu-2 released about
300 $\mu$M of dissolved Fe at pH 2. The dissolving solution (200 mL) was sampled
after filtration (0.2 $\mu$m polycarbonate filter). The clear solution was subjected to
changing acidity from pH 2 to 5 by the stepwise addition of dilute $NH_4OH$. The
precipitated particles were separated out by 0.2 $\mu$m filters and were used in TEM
and Mössbauer analysis. Size distributions for the Fe-bearing particles formed in
the suspensions were determined by a Horiba LB-500 light scattering microscopy
within the size range of 3-6000 nm.
**3 Results and discussion**
**3.1 Characterization of mineral samples**
The characteristic results are shown in Table S1 and Table S2. The samples
exhibited $S_{BET}$ in the range from $4.3 \pm 0.3$ to $22.6 \pm 2.3$ m$^2$/g. The $Fe_T$ content were
$5.45 \pm 0.34\%$, $26.30 \pm 0.57\%$, $2.36 \pm 0.56\%$ and $1.48 \pm 0.56\%$, for IMt-2, NAu-2,



SWy-2 and ATD, respectively. The proportions of $Fe_2O_3$ in IMt-2, NAu-2, SWy-2
and ATD were 7.95%, 39.03%, 5.55% and 2.57%, respectively.

**3.2 Effect of simulated CP on heterogeneous transformation of $SO_2$**

The *in situ* DRIFTS spectra on the IMt-2, NAu-2, SWy-2 and ATD samples before
and after CP exposed to $SO_2$ as a function of time are shown in Figure 1. For the
IMt-2 sample before CP (Figure 1a and 1b), the intensities of the broad peaks from
3600 to 3000 $cm^{-1}$ and a weak peak at 1650 $cm^{-1}$ increased with time. The band
between 3600 and 3000 $cm^{-1}$ was attributed to the vibrations of hydrogen-bonded
hydroxyl species (Zhao et al., 2015), while the absorption peak at 1650 $cm^{-1}$ was
mainly associated to $H_2O$ produced from the reaction between $SO_2$ and surface
hydroxyls (Nanayakkara et al., 2012; Cheng et al., 2016). A weak vibration was
observed at around 1100 $cm^{-1}$, which might be attributed to free sulfate anions on the
particle surface (Ullerstam et al., 2003; Nanayakkara et al., 2012; Yang et al., 2016).
Previous studies established that various types of surface OH groups are the key
reactive sites for sulfite/sulfate and bisulfite/bisulfate formation on mineral oxides
(Faust et al., 1989; Usher et al., 2003; Ullerstam et al., 2003), because of the
complexes formed between sulfite/sulfate species and the surface OH. Generally, the
$SO_2$ adsorption grow in intensity with decreasing OH stretching and $H_2O$ banding
(Zhang et al., 2006). However, the OH peaks herein were not observed to decrease
with prolonged time, because the losses of $H_2O$ and OH groups on the particle
surfaces were replenished by maintaining the constant RH in this study.
When the same set of experiments were carried out using the IMt-2 sample after CP
(Figure 1b), the intensities of the prominent peaks were significantly higher than those
on the IMt-2 sample before CP. Four new bands were readily observed at 1167, 1100,
1088 and 1077 $cm^{-1}$. The new bands were easily assigned to the stretching motion of



surface-coordinated sulfate species (1167 cm$^{-1}$), i.e., bidentate surface sulfate
complexes, free sulfate ion (1100 cm$^{-1}$), and sulfite/bisulfite species (1088 and 1077
cm$^{-1}$) (Peak et al., 1999; Ullerstam et al., 2003; Yang et al., 2016). These new bands
remained when an argon blow-off process was carried out, suggesting that the
surface-adsorbed sulfite/sulfate species between 1250 and 1000 cm$^{-1}$ was
chemisorbed (Zhang et al., 2006).

Upon adsorption of $SO_2$ on the surface of the NAu-2 sample before CP (Figure 1c

and 1d), the broad band from 3600 to 2800 cm$^{-1}$ and the peaks at 1580 and 1675 cm$^{-1}$
increased drastically with time. These absorbance bands were all attributed to the
surface hydroxyl species (OH) and $H_2O$. No peaks were observed over the range of
1000 to 1250 cm$^{-1}$, suggesting that the sulfite/sulfate products were not formed newly
on the surface of the NAu-2 sample before CP. Upon adsorption of $SO_2$ on the surface
of the NAu-2 sample after CP (Figure 1d), the new bands at 3661 and 3450 cm$^{-1}$, the
broad band between 3400 and 2700 cm$^{-1}$, and the broad band centered at 2131 cm$^{-1}$,
were observed as the exposure time increased. In detail, the band at 3661 cm$^{-1}$ could
be assigned to stretching vibration modes of isolated or bridged surface hydroxyl
groups bonded to the surface iron ions embedded in the octahedral and tetrahedral
sites (Faust et al., 1989; Nanayakkara et al., 2012; Zhao et al., 2015). The peaks at
around 3450 cm$^{-1}$, 2131 cm$^{-1}$ and the band between 3400 and 2700 cm$^{-1}$ were all
attributed to surface OH groups (Ma et al., 2010; Zhao et al., 2017). These new bands
generated on the processed NAu-2 particles suggested that CP changed the location of
diverse OH groups on the particle surfaces. Over the range of 1250-1000 cm$^{-1}$, the
new bands centered at 1170 cm$^{-1}$ was assigned to the asymmetric stretching of sulfate
(Kong et al., 2014; Yang et al., 2015).



The spectra of the SWy-2 samples before and after CP (Figure 1e and f) showed a
similar spectral character with those of the NAu-2 samples. The spectra for the ATD
samples before and after CP (Figure 1 g and h) were roughly the same as the ones for
IMt-2. All of the results demonstrated that the characteristic peaks for the active OH
sites and the sulfite/sulfate products on the mineral particles after CP were
significantly higher than those on the ones before CP, indicating the higher
hygroscopicity and more $SO_2$ uptake on the particles after CP. The data shown herein
confirmed that CP could potentially promote the transformation of $SO_2$ on the particle
surfaces.
**Figure 1**
**3.3 Uptake coefficient of $SO_2$ on the mineral particles before and after CP**
The areas of the bands (from 1250 to 1000 $cm^{-1}$) attributed to the sulfite/sulfate
products as a function of time are shown in Figure 2. It was evident that the peak
areas of the products on the mineral particles after CP were generally greater than the
ones before CP. The reaction on the sample surfaces was practically saturated to $SO_2$
uptake within 15 min, except for the NAu-2 and IMt-2 samples after CP. As for all of
the sample, the saturation coverages of the sulfite/sulfate products after CP were
obviously greater than the corresponding values before CP, suggesting that CP
favored the sulfate formation on the mineral surfaces due to improving active site
number, as expected previously.
**Figure 2**
The maximum uptake coefficients ($\gamma_{geo}$ and $\gamma_{BET}$) for $SO_2$ uptake on the samples
were estimated on the basis of the sulfate formation rates in the initial 15 min. The
values on the mineral samples before and after CP are shown in Table 1. The $\gamma_{geo}$
values of $SO_2$ on the IMt-2, NAu-2, SWy-2 and ATD samples before CP were



$1.03\times10^{-7}$, $0.30\times10^{-7}$, $1.72\times10^{-7}$ and $1.37\times10^{-7}$, respectively, which were in the order
of SWy-2, ATD, IMt-2 and NAu-2. The $\gamma_{geo}$ values of $SO_2$ on the IMt-2, NAu-2,
SWy-2 and ATD samples after CP were 4.7, 19.4, 2.7 and 2.0 times higher than the
values before CP, respectively, suggesting that the $SO_2$ uptake on the mineral particles
significantly increased after CP.

$A_{BET}$ was more appropriate to represent the effective area, because the reactant may

diffuse into tiny holes of the entire sample. The $\gamma_{BET}$ values of $SO_2$ on the IMt-2,
NAu-2, SWy-2 and ATD samples before CP were $2.62\times10^{-12}$, $0.75\times10^{-12}$, $3.70\times10^{-12}$
and $1.61\times10^{-11}$, respectively, which were in the order of ATD, SWy-2, IMt-2 and
NAu-2. It was noteworthy that the $S_{BET}$ of samples increased after CP, as shown in
Table 1. The $\gamma_{BET}$ values of $SO_2$ on the IMt-2, NAu-2, SWy-2 and ATD after CP were
2.2, 4.1, 1.5 and 1.4 times higher than the values before CP, respectively. The
discrepancies in the $\gamma_{BET}$ value confirmed that the higher sulfate formation rates of the
particles after CP was not only due to the increased surface area of the particles, but
also resulting from the chemical modification on the particle surfaces.

The estimated uptake coefficients were several orders of magnitude lower than the

results from Ullerstam et al. (2003) and Usher et al. (2003), which could be partly
explained by the difference in the preparation of mineral dust samples, or the
difference between diverse experimental structures such as the DRIFTS and Knudsen
cell in kinetics discussion. In this study, mineral dust particles were in a highly
accumulative state in the sample support of Knudsen cell. The many layers of
particles in the latter study will hinder the diffusion of gas into the underlayer
particles, resulting in the underestimate of $\gamma_{BET}$. However, the values herein were
comparable to those obtained by the similar DRIFTS setup (Fu et al., 2007),
indicating the reliability of our measurements.





In addition, the formation rate of sulfate appeared a linear increasing trend as a
function of pH cycles. Specifically, the increasing amount of sulfate ions for the
IMt-2, NAu-2, SWy-2 and ATD samples after each pH cycle during CP were $7.0\times10^{10}$,
$1.0\times10^{11}$, $5.0\times10^{10}$, $3.0\times10^{10}$, in the order of NAu-2 > IMt-2 > SWy-2 > ATD (Figure
3). The $\gamma_{BET}$ ($\gamma_{geo}$) for IMt-2, NAu-2, SWy-2 and ATD after CP were 2.2 (4.7), 4.1
(19.4), 1.5 (2.7) and 1.4 (2.0) times greater than the corresponding values for those
without CP procedure, respectively. The multiples factors for $\gamma_{BET}$ ($\gamma_{geo}$) were
coincided with the total Fe content of these samples: NAu-2 (26.30%) > IMt-2
(5.45%) > SWy-2 (2.36%) > ATD (1.48%). We thus supposed that the $SO_2$ uptake on
these dust samples were closely related to the Fe hosted in the particles.
**Figure 3**
**3.4 Morphological change of the mineral particles after CP.**
Figure 4 shows the TEM images of the mineral particles before and after CP. As
shown in Figure 4 a, c, e and g, the IMt-2, NAu-2, SWy-2 and ATD samples before
CP primarily consisted of laminar aluminosilicate with irregular shape and rough
morphologies mainly at the micrometer scale, all of which were characterized by
various fractions of Fe (1.5%-26.2%), along with minor Mg (0.1%-16.5%), K
(0.0%-7.8%) and Ca (0.0%-1.1%). The Fe within the aluminosilicates of the particles
was evenly distributed. Besides, some Fe-rich crystal with several hundreds of
nanometers in size were found to attach onto the ATD particles, which were identified
as $\alpha$-$Fe_2O_3$ (PDF: 33-664) from the typical $d$-spacing analysis of HRTEM (Janney et
al., 2000).
After the simulated CP, all of the processed mineral particles showed much smaller
size than the ones before CP. For example, the typical IMt-2 and NAu-2 particles after
CP (Figure 4 b and d) were < 1 $\mu$m in size. Under the TEM, the average Fe content of





the individual IMt-2 and SWy-2 particles (Figure 4 b and f) decreased from 5.5% (±
1.9%; $n = 50$) to 4.1% (± 1.6%; $n = 50$) and from 2.4% (± 0.6%; $n = 50$) to 2.1% (±
0.5%; $n = 50$), respectively. In addition, the IMt-2 particles after CP showed a
heterogeneous distribution of the Fe on the basis of the EDX data. Most of the
aluminosilicate in IMt-2 after CP hosted lower Fe content (4.1%), whereas a few of
the Fe-rich particles with less Si/Al were observed with irregular shapes at the
nanoscale. The TEM images of the NAu-2 and ATD particles after CP (Figure 4 h)
showed some pseudohexagonal nanoparticles with around 5 nm in diameter. Based on
the EDX and SAED analysis, these nanoparticles were Fe-rich and the $d$-spacings was
at about 1.5-2.5 Å, all of which were identified to be 2-line ferrihydrite (Janney et al.,
2000; Shi et al., 2009).
The TEM observation suggested that CP induced the disintegration of mineral
particles and thus produced enhanced surface area, resulting in more active sites
available on the particle surfaces for $SO_2$ uptake. Results of TEM also showed that CP
influenced the Fe mineralogy, and lead to the Fe-rich nanoparticle formation, which
could partly explain the higher $SO_2$ uptake on the mineral particles after CP.
**Figure 4**
**3.5 Fe speciation analysis before and after CP.**
The fractions of "free-Fe" (including $Fe_A$ and $Fe_D$) and "structural-Fe" in the
mineral particles before and after CP were determined by the CBD extraction (Figure
5). In terms of total Fe, the amorphous Fe ($Fe_A$) (e.g., nanoparticulate and poorly
crystalline ferrihydrite) contents of the IMt-2, NAu-2, SWy-2 and ATD samples
before CP were 0.7%, 0.5%, 0.7% and 3.8%, respectively. The crystalline Fe
(oxyhydr)oxides ($Fe_D$) (e.g., α-FeOOH and α-$Fe_2O_3$) contents of the IMt-2, NAu-2,
SWy-2 and ATD samples before CP were 7.2%, 2.3%, 4.5% and 35.5%, respectively.



As a result, the fractions of "structural-Fe" before CP were 92.1%, 97.2%, 94.8% and
60.7%, respectively, for IMt-2, NAu-2, SWy-2 and ATD.
After CP, the $Fe_A$ contents of the IMt-2, NAu-2, SWy-2 and ATD samples reached
1.8%, 1.2%, 1.7% and 24.2%, respectively, which increased by 2.6, 2.4, 2.4 and 6.4
times as compared to the ones before CP. The crystalline Fe (oxyhydr)oxides ($Fe_D$)
contents of the samples after CP were not significantly changed as compared to the
ones before CP; whereas the content of "structural-Fe" in the Al-Si crystals of the
IMt-2, NAu-2, SWy-2 and ATD samples after CP decreased by various degrees, to
91.1%, 96.1%, 93.2% and 42.5%, respectively. We thus proposed that the increased
fractions of $Fe_A$ could be mostly transformed from the "structural-Fe" in the
aluminosilicate phase of the particles during CP, which is in good agreement with the
TEM observation. For example, the $Fe_A$ in the ATD samples increased from 3.8% to
24.2% after CP, accompanied by a sharp decrease of the structural-Fe content from
60.7% to 42.5%.
**Figure 5**
The Mössbauer spectra and their fitted results are shown in Figure 6. The
corresponding hyperfine parameters estimated from the best fitted spectra are
presented in Table S3. The central doublet with isomer shift (IS) of 0.37 mm s$^{-1}$ and
quadrupole shift (QS) of 0.72 mm s$^{-1}$ were typical for high-spin Fe(III) in octahedral
symmetry (Eyre and Dickson, 1995), while the other one with IS of 1.12 mm s$^{-1}$ and
QS of 2.65 mm s$^{-1}$ was characteristic of high spin Fe(II) (Hofstetter et al., 2003;
Kopcewicz et al., 2015). The two doublet components of the IMt-2, NAu-2, SWy-2
and ATD samples before CP were all attributed to different fractions of Fe(III) and
Fe(II) in the aluminosilicate crystals, respectively. Before CP, the Fe(II) fraction in the
IMt-2, NAu-2, SWy-2 and ATD samples were 34.0 %, 12.9 %, 18.3 % and 29.0 %,





respectively (Figure 6 a, c, e and g). Furthermore, the spectra of the ATD sample
before CP showed not only two central quadrupole doublets, but also one MHS sextet
with IS of 0.39 mm s$^{-1}$, QS of $-0.13$ mm s$^{-1}$ and H$_f$ of 51.1 T. The MHS sextet, which
shared 31.8 % of the whole area, could be ascribed to α-Fe$_2$O$_3$ (Kopcewicz and
Kopcewicz et al., 1991), in agreement with the TEM analysis and "free-Fe"
measurement as mentioned previously.
After CP, the Fe(II) content of the samples decreased to 31.5 %, 11.6 %, 17.1% and
10.9%, respectively, for IMt-2, NAu-2, SWy-2 and ATD (Figure 6 b, d, f and h). It
was supposed that the Fe(II) release is more energetically favorable than one of Fe(III)
due to the bond strength. As to the ATD sample after CP (Figure 6 h), not only did the
Fe(II) fraction decrease from 29.0% to 10.9%, but also the Fe(III) fraction in the
aluminosilicates decreased from 39.0% to 33.0%. Meanwhile, the α-Fe$_2$O$_3$ fraction
was not significantly changed (31.8% vs. 32.3%). As discussed previously, the Fe
mobilization was dependent on the specific chemical bonds. The Fe$_D$ phase in
α-Fe$_2$O$_3$ with the strong Fe−O bond was less liable than that embedded in the
aluminosilicate lattice (Strehlau et al., 2017). It was well documented that the Fe
replacing alkaline elements as the interlayer ions was easy to be mobilized than the Fe
bound by covalent bonds in the aluminosilicate matrix (Luo et al., 2005; Cwiertny et
al., 2008; Journet et al., 2008). Therefore, the Fe in the aluminosilicate fraction of the
mineral particles exhibited varied iron solubility.
Particularly, a new quadrupole doublet with IS of 0.67 mm s$^{-1}$ and QS of 1.21 mm
s$^{-1}$ was observed in the spectra of the ATD sample after CP (Figure 6 h), which shared
23.8% of the total area, and was possibly indicative of the Fe(III) oxide hybridized in
the aluminosilicate matrix (Kopcewicz and Kopcewicz, 1991). The "free-Fe"
measurement have indicated that the Fe$_A$ fraction of ATD increased by 20.4% after CP,





so that this Fe phase was most likely to be amorphous Fe(III) hybridized with Al/Si.
In the terms of the other samples after CP, the magnetic signal of the newly formed
Fe(III) phase was not detected. It was probably due to the newly formed Fe fractions
were not available at sufficiently high level to be clearly resolved by the Mössbauer
spectroscopy, and/or the slight signal drift and the poor signal to noise ratio made an
unambiguous identification difficult. Herein, the newly formed amorphous Fe(III)
phase was supposed to be a reactive Fe-bearing component, of which may contribute
significantly to the $SO_2$ uptake even at a low level.
**Figure 6**
**3.6 The dissolution-precipitation cycle of the mineral Fe during CP**

During the simulated CP experiments, the concentrations of total dissolved Fe ($Fe_s$),

dissolved Fe(II) and Fe(III) released from the particles as a function of time are shown
in Figure 7. Similar dissolution trends were observed for all of the samples. One can
see that the suspensions at pH 2 induced a rapid increase of $Fe_s$. Once increasing the
pH from 2 to 5 resulted in a rapid and almost complete removal of $Fe_s$. In fact, only a
rather small fraction of the Fe in dusts could be dissolved at pH above 4 (Zuo and
Hoigne, 1992). The dissolved Fe precipitated rapidly as insoluble deposit at pH 5.
When the suspension pH was again reduced to 2, a steep increase in the $Fe_s$
concentration was measured once again. The fast Fe release was due to the
redissolution of the Fe-rich precipitates, which was proposed to be reactive Fe phases.
Such highly soluble Fe-bearing precipitates have been observed under the TEM, as
well as the "free Fe" measurement and Mössbauer characterization.
**Figure 7**

For each pH cycle during the simulated CP experiment, the overall changes of total

released Fe concentrations were reproducible. The Fe ion on the particle surfaces



would experience a continuous dissolution-precipitation-redissolution-reprecipitation
process when the pH cycles between pH 2 and pH 5 (cloud-aerosol modes). During
this process, the Fe(II) fraction would be transformed to Fe(III). The results shown
herein suggested that CP could significantly modify Fe partitioning between dissolved
and particulate phases in the real atmosphere. Not only did the increase of specific
surface area contribute to the enhanced sulfate formation, but also the highly reactive
Fe on the particle surfaces yielded during CP were also responsible for the higher $SO_2$
uptake on the particles after CP.
When investigating the NAu-2 sample, once the pH of the clear solution increased
from 2 to 5-6, the Fe-bearing nanoparticles separated out from the solution rapidly
and precipitate out slowly. It developed an initial yellow color and then an orange
colored suspension. The TEM images of the precipitated particles are shown in Figure
8. The particles could be categorized into two different types. One type of particle
could be characterized as hundreds of nanometers in size, with low Fe but high Si/Al
content. The other type displayed particle sizes nearly 1 micrometer, and were Fe-rich
but contained a smaller amount of Si/Al components. These bigger particles were
ambiguously identified as $Na_{0.42}Fe_3Al_6B_{309}Si_6O_{18}(OH)_{3.65}$ (PDF: 89-6506) on the
basis of the EDX data and SAED analysis. It is likely that the Al/Si elements also
precipitated out along with the Fe.
**Figure 8**
The Mössbauer spectra of the precipitated Fe-rich particles are shown in Figure 9.
Two central doublets were distinguished, with one (48.4%) of IS = 0.45 mm $s^{-1}$, QS =
0.75 mm $s^{-1}$, and the other (51.6%) of IS = 0.24 mm $s^{-1}$, QS = 0.76 mm $s^{-1}$. Both of
the two doublet components could be attributed to the Fe(III) fraction in the
aluminosilicates (Kopcewicz et al., 2015). The results were in good agreement with





the TEM observation, which showed that most of these Fe particles were mostly
present as the Fe(III) hybridized with Al/Si. The particle size distributions in the
suspensions were also determined by dynamic light scattering, as shown in Figure 10.
When pH was lower than 2.0, the particles seemed to stabilize below 10 nm in size.
These Fe colloids were thought to be a source of soluble Fe (Janney et al., 2000).
Once pH increased, the size of precipitated particles quickly increased, even to
micro-scale, and the suspension was featured with a polydispersed size distribution.
Conclusively, the precipitated Fe were mainly Fe(III) with weak crystal structure
and/or ferrihydrite nanoparticle hybridized with Al/Si, which were possibly
transformed from the Fe hosted in the aluminosilicate matrix of the particles. The
particle surfaces after CP was coated by these reactive Fe, resulting in enhanced $SO_2$
uptake.
**Figure 9**
**Figure 10**
**4 Conclusion and implication**
Transition metal ions, especially Fe(III), could catalyze $SO_2$ oxidation rapidly in
cloud drops (Harris et al., 2013). This study further confirmed that $SO_2$ uptake on the
mineral particles could be greatly enhanced during CP, possibly more than described
previously. The higher uptake coefficient of the particles after CP was not only due to
increased surface area, but also resulted from the chemical modification of the particle
surfaces. The "free-Fe" and Mössbauer analysis suggested that CP triggered newly
formation of amorphous Fe particles on the surfaces, of which were mostly
transformed from the Fe hosted in the aluminosilicate matrix. TEM showed that the
amorphous Fe(III) and/or ferrihydrite nanoparticle were hybridized with Al/Si. In
general, the acidity fluctuation during CP enables the dissolution-precipitation cycles





of mineral Fe to yielded more reactive Fe, resulting in more $SO_2$ uptake on the
particle surfaces. More $SO_2$ adsorption further increases the surface acidity of dust
particles, in turn leading to higher Fe solubility; again, more sulfate formation. It was
thus proposed that there is a positive feedback relative to $SO_2$ update and iron
mobilized from mineral particles during CP, therefore enhanced sulfate formation
greatly.

Our results also serve to explain high sulfate loading in fog-haze episodes of China.

It has been recommended that sulfate contributed significantly to the explosive
growth of fine particles, thus exacerbating severe fog-haze development (Kasibhatla
et al., 1997; Nie et al., 2014; Barrie et al., 2016). Haze and fog within an episode was
often found to transform each other at a short time due to the diurnal variation of RH,
whereby the haze-fog transition was probably analogous to the aerosol-cloud
interaction. Water content of aerosol or fog drops was regulated by RH, and thus
allowed the particle acidity fluctuation. Although the aerosol acidity could not be
accurately determined from field measurements or calculated using the
thermodynamic model, we recognized that the large pH fluctuations between the
haze-fog modes could significantly modify the microphysical properties of mineral
particles, and triggered formation of reactive Fe particles and thus accelerated sulfate
formation via a self-amplifying process, contributing to explosive growth of fine
particles at the initial stage of fog-haze events. The data presented herein also
highlight that CP provide more bioavailable iron from mineral particle than one
expected previously, of which is a key speciation to promote oceanic primary
productivity. Results of this study could partly explain the missing source of sulfate
and improve agreement between models and field observations.



Additionally, previous studies indicated that Fe in pyrogenic aerosols was always
presented as liable Fe, such as ferric sulfate and aggregated nanocrystals of magnetite
($Fe_3O_4$) (Fu et al., 2012), and displayed higher Fe solubility compared to dust
(Desboeufs et al., 2005; Sedwick et al., 2007; Ito et al., 2019). Alexander et al.
demonstrated that the sulfate formed through metal catalysis was highest over the
polluted industrial regions of northern Eurasia, suggesting that the increasing
importance of the metal-catalyzed S(IV) oxidation pathway due to anthropogenic
emissions (Alexander et al, 2009). With the rapid development of industry and
agriculture, the pyrogenic Fe-containing aerosols are indispensable contributors to the
atmospheric Fe load in China. Thus, the acidic solution at pH 2 and high sulfate
loading of fine particles in severe fog-haze events of China might be more relevant to
Fe-containing combustion aerosols than mineral dust. Based on the current findings,
not only the potential influences of cloud liquid water content, light, and organic
ligands, but also the solubility and speciation of Fe in pyrogenic aerosols will be
considered during the simulated CP experiments in the future. A more detailed
understanding of the iron-sulfur cycle during CP is therefore critical to estimate
accurately the contribution of CP to global sulfate loading and its impact on the
climate.

*Author contributions.* Z.W., H.F. and J.C. designed the experiments, Z.W., T.W., H.F. and L.Z. performed the laboratory experiments. H.F., J.C., L.Z. and V.G. contributed reagents/analytic tools. C.G., V.G. and M.T. gave some valuable suggestions in



designing the experiments. Z.W., T.W. and H.F. analyzed data. Z.W. and H.F. wrote the manuscript, with inputs from all coauthors.

*Competing interests*. The authors declare no conflict of interest.

*Acknowledgments*. This work was supported by National Key R&D Program of China (2016YFC0202700), National Natural Science Foundation of China (Nos. 91744205, 21777025, 21577022, 21177026), and International Cooperation Project of Shanghai Municipal Government (15520711200) and Opening Project of Shanghai Key Laboratory of Atmospheric Particle Pollution and Prevention.

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


# Captions of Figures and Tables

**Figure 1.** Comparison of the DRIFT spectra of mineral dust samples upon exposure to $SO_2$ for 45 min before and after CP. Data for IMt-2 (a and b), NAu-2 (c and d), SWy-2 (e and f), ATD (g and h), are shown in the ranges of 4000 to 1250 $cm^{-1}$ and 1250 to 1000 $cm^{-1}$, respectively.

**Figure 2.** Comparison of the integrated areas on DRIFTS spectra in the range of 1250-1000 $cm^{-1}$ for the sulfate species formed on the samples before and after CP.

**Table 1.** Sulfate formation rates and uptake coefficients of $SO_2$ on particle samples before and after CP.

**Figure 3.** Comparison of the sulfate formation rates as a function of pH cycle.

**Figure 4.** Comparison of morphologies and chemical properties for samples collected before and after CP using TEM. The dotted circles indicate the positions of the electron beam for the HRTEM images and SAED patterns. Elements of the detected parts of individual particles are also presented. Square brackets indicate mass percent of iron. The iron species were identified by the Miller indices and the SAED patterns. (a) IMt-2 particles characterized by high fractions of Al and Si, along with other crustal elements including Mg, K and Fe. (b) IMt-2 particles after CP were almost all less than 1$\mu$m in size. Some Fe-rich particles with less Si and Al were observed on nanoscale dimension. (c) NAu-2 particles with high Fe/Si-ratios, contain Mg, Al and Ca elements. (d) NAu-2 particles after CP were much smaller than the ones before CP. Some ferrihydrite clusters were observed and were attached on the surface of the NAu-2 particles after CP. (e) Typical SWy-2 particles were Al/Si-rich, containing Fe, Mg and Ca elements. (f) TEM images of the SWy-2 particles after CP appeared smaller than the particles before CP. (g) The Si/Al-rich crystal in ATD particles was



aluminosilicate with low content of Fe, and a typical of the α-Fe$_2$O$_3$ particles (PDF: 33-664) was found to attach onto the aluminosilicate surface. (h) The pseudohexagonal nanoparticles were observed to on the surface of α-Fe$_2$O$_3$ crystal among the ATD particles. The SAED lattice constant of these nanoparticles were found to be very close to that of 2-line ferrihydrite.

**Figure 5.** The fractions of "free-Fe" (Fe$_A$ and Fe$_D$) and "structural-Fe" were measured by the chemical CBD extractions for the samples before and after CP. Results are present as relative percentage of Fe$_T$.

**Figure 6.** Mössbauer spectroscopy measured for samples. IMt-2 before and after CP (a and b), NAu-2 before and after CP (c and d), SWy-2 before and after CP (e and f), ATD before and after CP (g and h). Experimental data were fit using a least-squares fitting-program. The IS values were relative to α-Fe at RT. Prominent spectral features associated with different iron species are indicated.

**Figure 7.** The concentrations of Fe$_s$, dissolved Fe(II) and Fe(III) in the suspensions measured over 144 h in the solution cycled between pH 2 and pH 5 for IMt-2 (a), NAu-2 (b), SWy-2 (c) and ATD (d), respectively.

**Figure 8.** TEM images of the newly formed particles in the precipitation experiment. Based on the TEM-EDX measurement and SAED analysis, these particles could be categorized into two different types, which were circled in Figure 8 a. The typical sizes of the first type were hundreds of nanometers. The enlarged images are displayed in Figure 8 b, c and d. The insert EDX data and SAED image confirmed that they were poor crystalline aluminosilicate with low Fe but high Si/Al content. The second type (Figure 8 e, f and g) were Fe-rich but with less amount of Si/Al, which were nearly 1 micrometer in size. Based on the EDX data and the SAED





analysis, these bigger particles were ambiguously identified as $Na_{0.42}Fe_3Al_6B_{309}Si_6O_{18}(OH)_{3.65}$ (PDF: 89-6506).

**Figure 9.** Mössbauer spectroscopy measured at RT for the neo-formed particles collected in the precipitation experiment.

**Figure 10.** During the precipitation experiment, the particle size distributions in the suspensions were determined by dynamic light scattering. The presented size distributions are characteristic of neo-formed nanoparticles or microparticles as the suspension pH raised from 1.0 to 3.8.



**Figure 1**

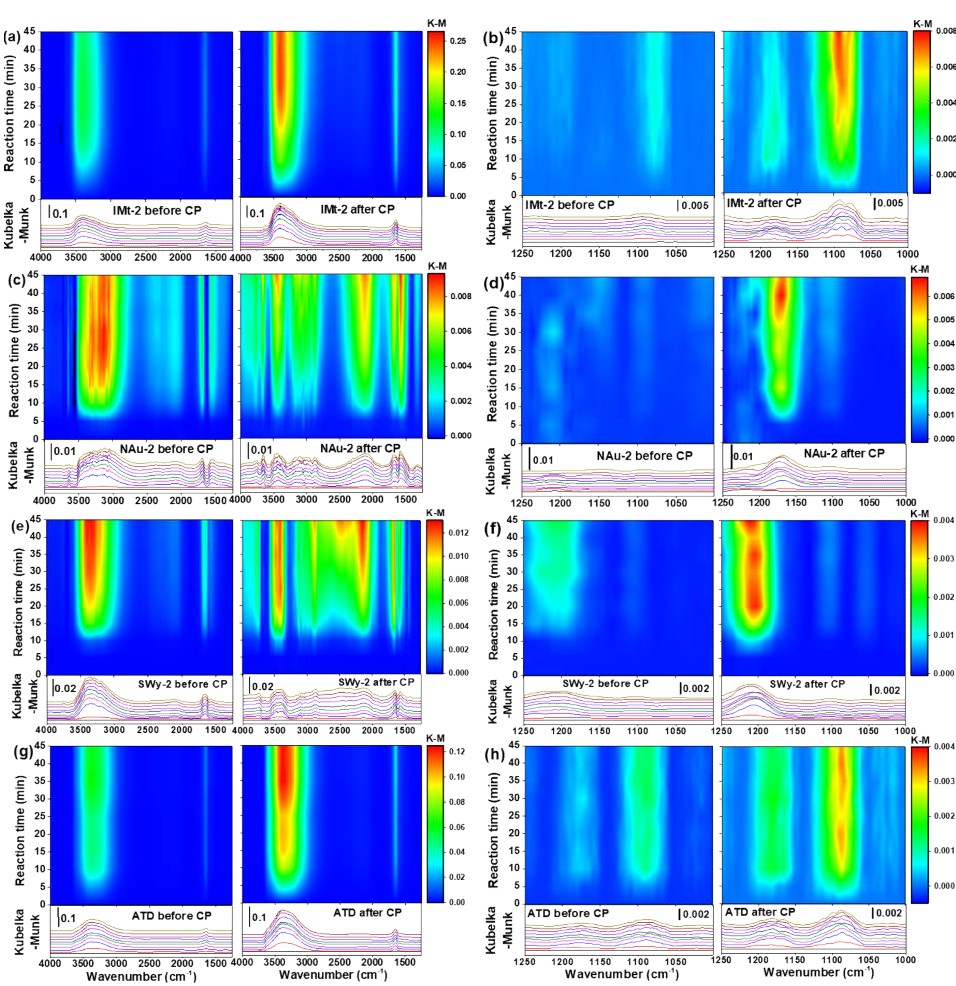





**Figure 2**

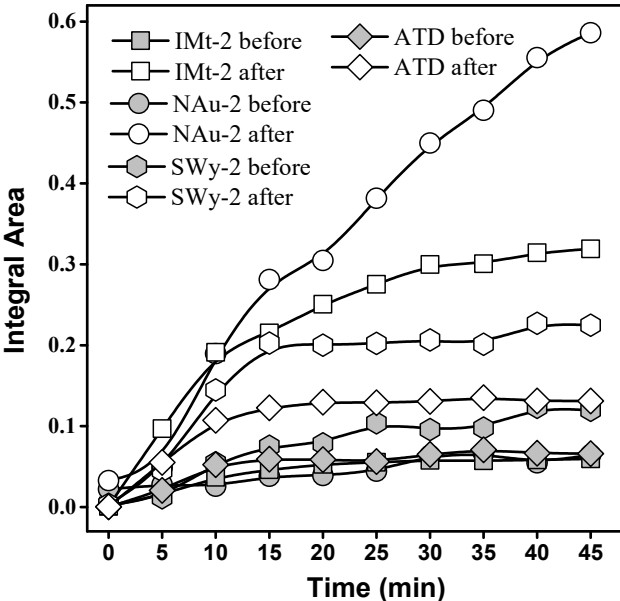



## Table 1

| Samples | $A_{BET}$ (m² g⁻¹) | Sulfate formation rate (ions s⁻¹) ($\times 10^{10}$) | $A_{geometric}$ (m²) ($\times 10^{-5}$) | $\gamma_{BET}$ ($\times 10^{-12}$) | $\gamma_{geometric}$ ($\times 10^{-7}$) |
|---|---|---|---|---|---|
| IMt-2 before CP | 20.1±1.5 | 6.13 | 1.95 | 2.62 | 1.03 |
| IMt-2 after CP | 32.0±2.6 | 28.72 | 1.95 | 5.76 | 4.85 |
| NAu-2 before CP | 19.8±1.3 | 1.80 | 1.95 | 0.75 | 0.30 |
| NAu-2 after CP | 93.7±7.5 | 34.57 | 1.95 | 3.06 | 5.83 |
| SWy-2 before CP | 22.6±2.3 | 10.20 | 1.95 | 3.70 | 1.72 |
| SWy-2 after CP | 40.8±1.5 | 27.19 | 1.95 | 5.49 | 4.59 |
| ATD before CP | 4.3±0.3 | 8.11 | 1.95 | 16.05 | 1.37 |
| ATD after CP | 6.5±1.0 | 16.33 | 1.95 | 22.33 | 2.76 |





**Figure 3**

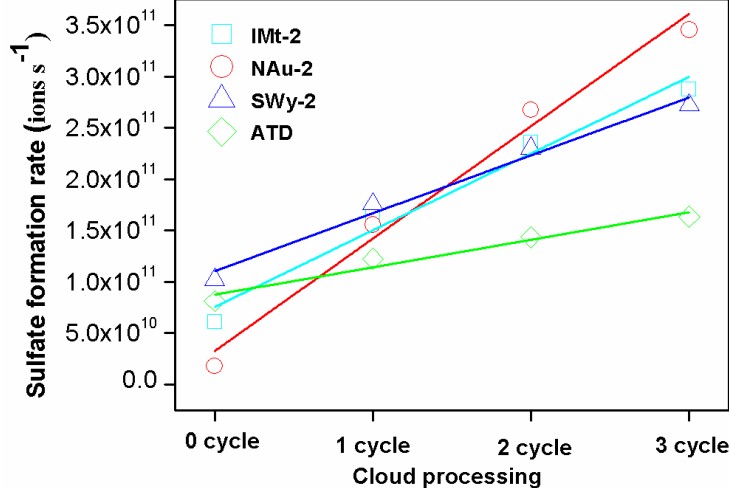





**Figure 4**

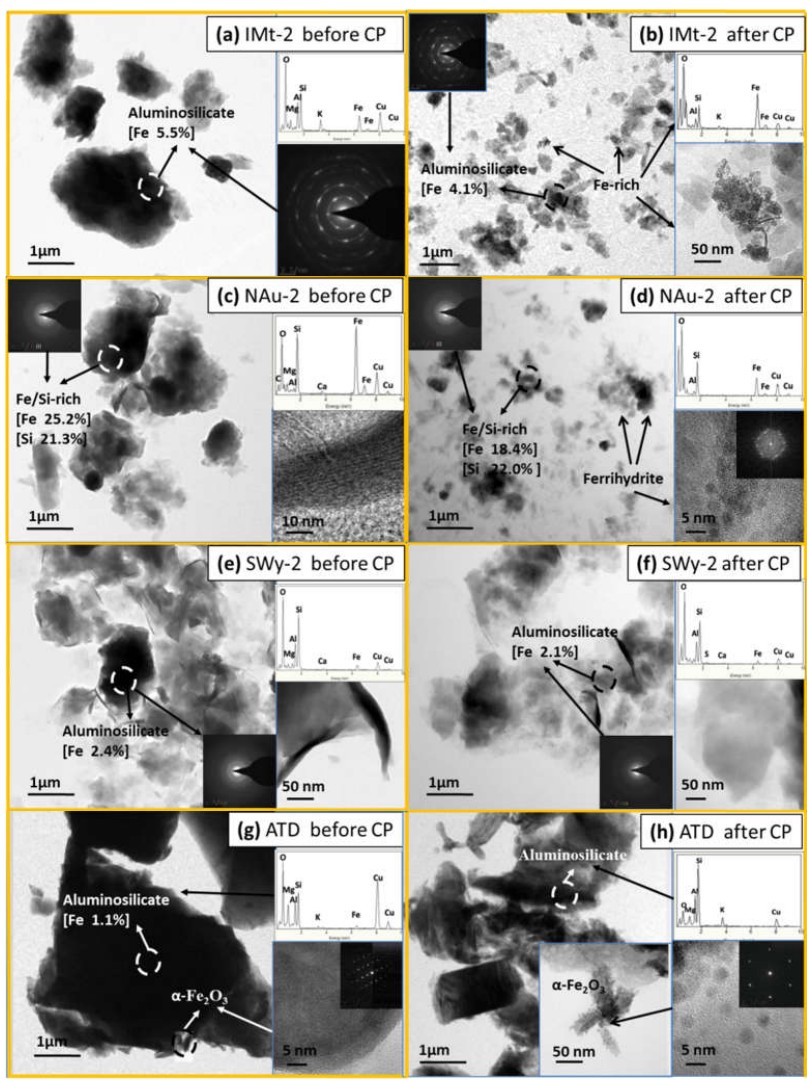



**Figure 5**

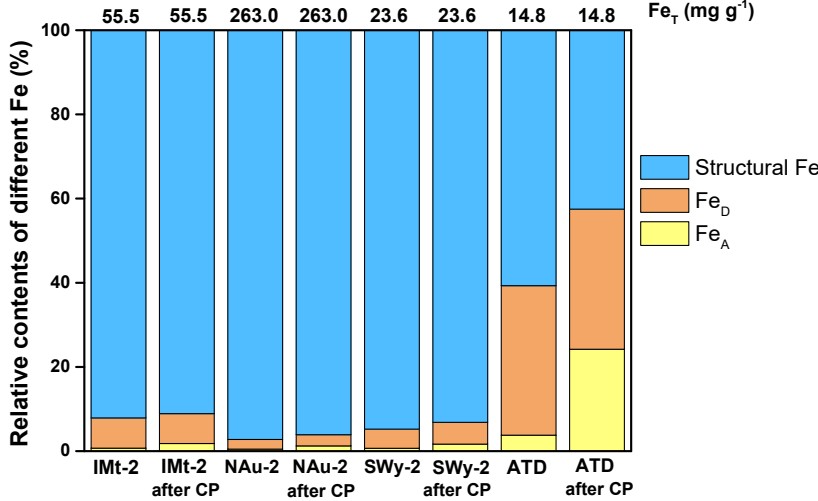



**Figure 6**

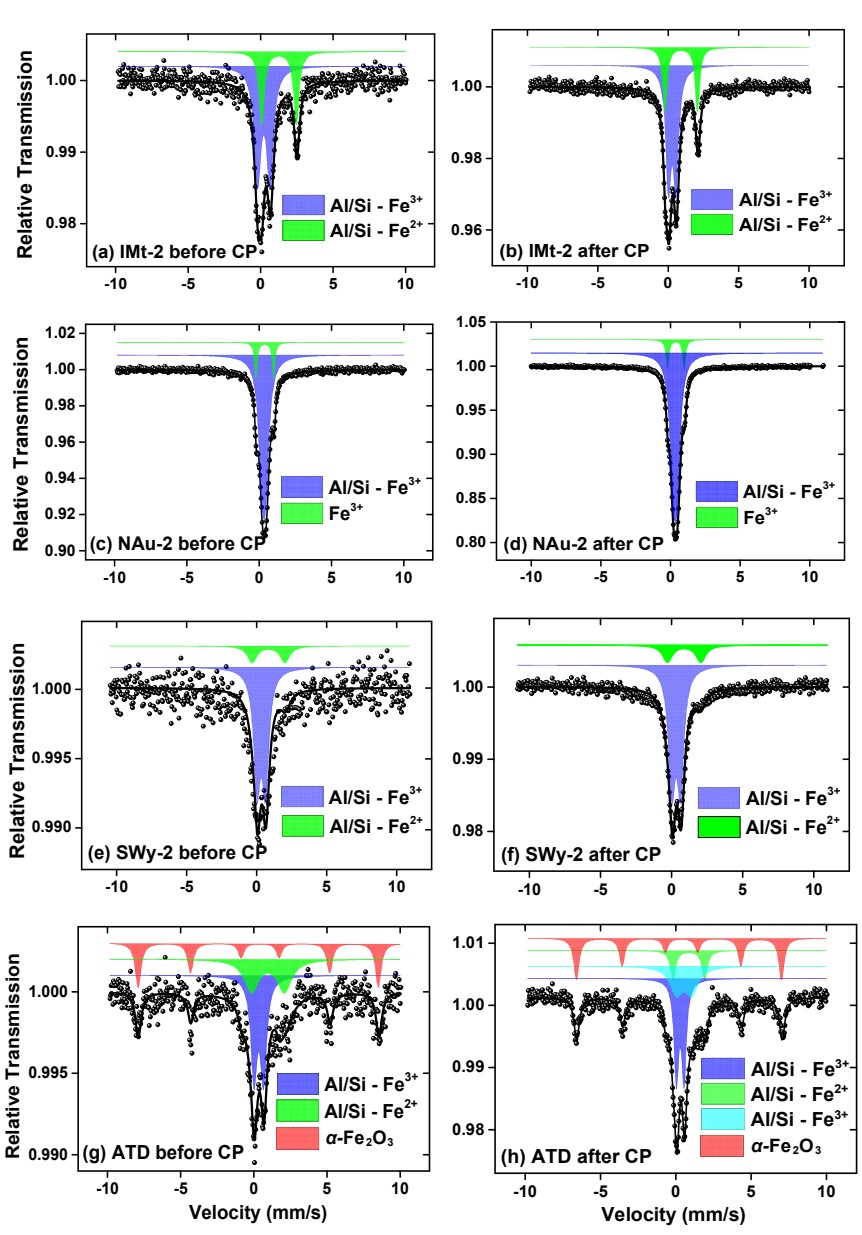





**Figure 7**

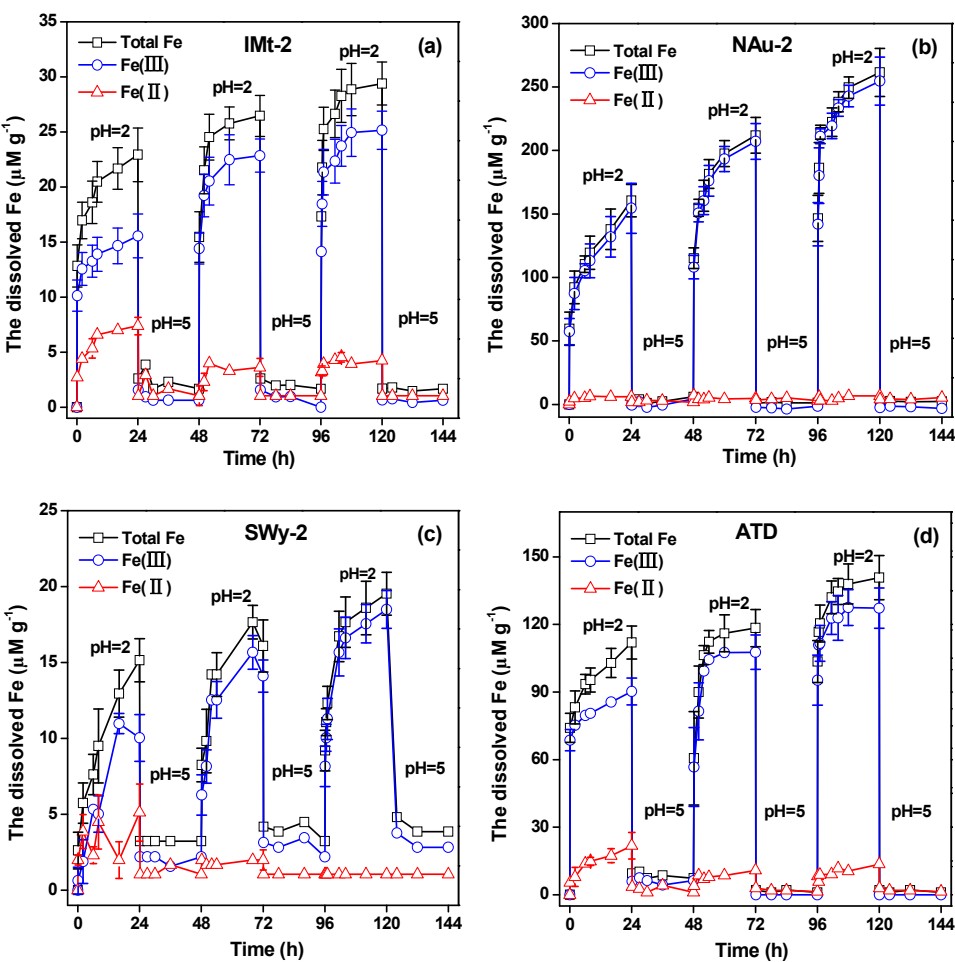



**Figure 8**

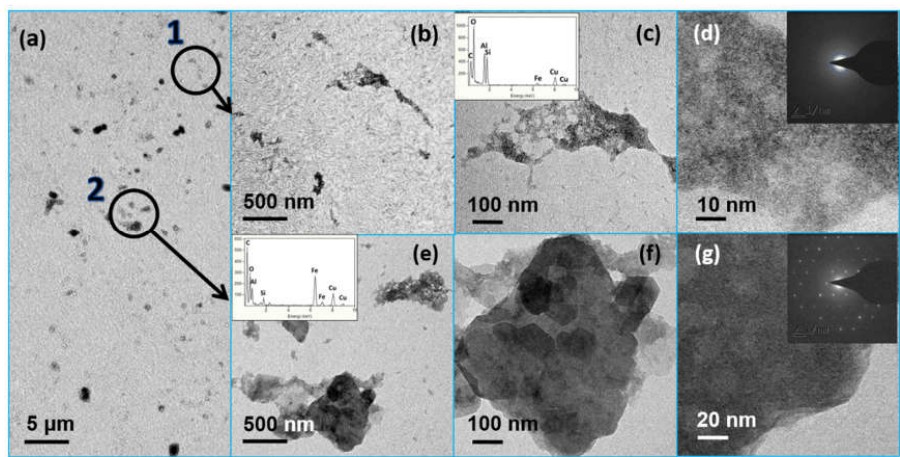


**Figure 9**

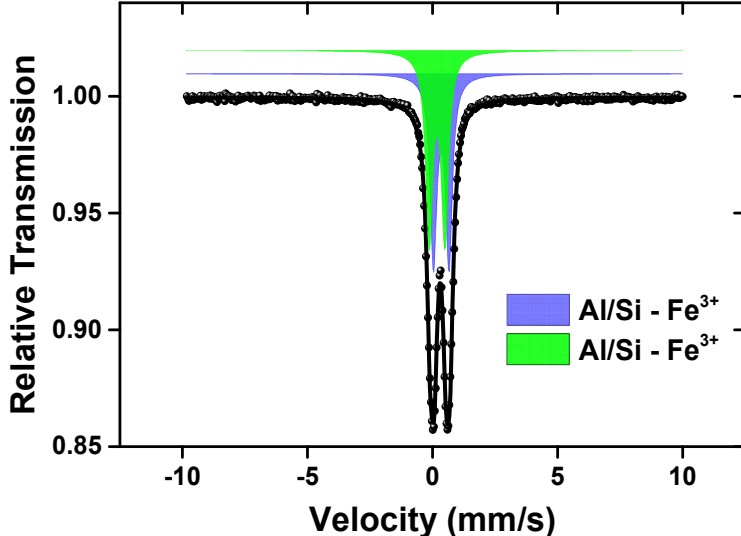





**Figure 10**

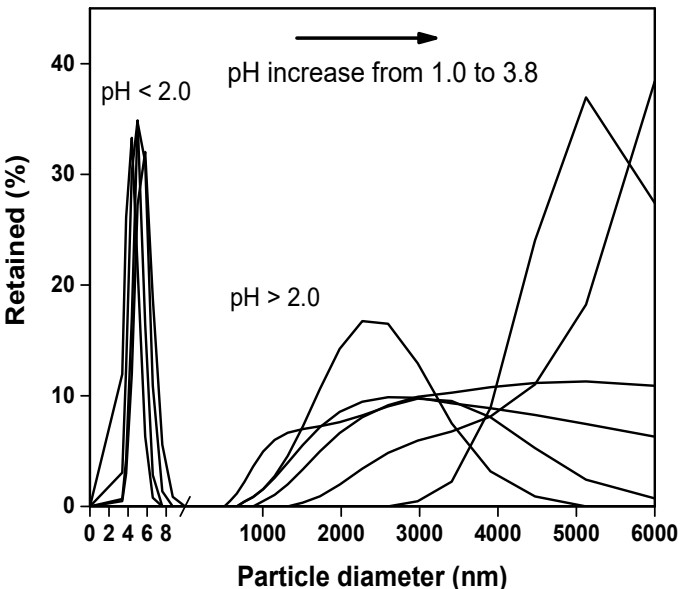