# Peer review of "Enhanced heterogeneous uptake of sulfur dioxide on mineral particles through modification of iron speciation during simulated cloud processing"

_Atmospheric Chemistry and Physics, 2019_

## Referee Comment (RC1) · Anonymous Referee #1 · 23 Jun 2019

The authors focused on the heterogeneous transformation of SO2 on mineral dust during cloud processing by the simulated experiment. They characterized Fe morphology using the combined methods including TEM, Mossbauer, and CBD extraction method. The iron mobilized from mineral particle was also measured. Generally, heterogeneous transformation on the surface of mineral particle play a vital role on second particle formation. The result shown in this manuscript shed a light on heterogeneous chemistry, and it is helpful to further understand the fog and haze formation in China. Thus, this manuscript made some new contribution to atmosphere chemistry. The

manuscript was well written and organized. The topic is of interest to the journal's readers. I thus recommended that this manuscript could be published on ACP. However, the manuscript also suffered from some flaws, and I listed as follows.

General questions/comments/suggestions: 1 The four examined clays were purchased from the clay mineral depository. The author should discuss why they were selected, particularly why they are representative of clays in aerosols.

2 Line 159: "TEM observation". The methods of TEM sample preparation will change the aggregation state of such aggregates. Some researchers have extensively worked on this and gave detailed information how they did it. Nothing is reported here on sample preparation. Therefore, please supply the detail about TEM sample.

3 The FeA and FeD content of Arizona Test Dust was "$0.067\pm 0.005\%$ for FeA and $0.41\pm 0.04\%$ for FeD" in Shi et al. (2011), which was not completely in accordance with the values in Table S5. The author should try to explain the discrepancy of FeA and FeD content.

Further specific comments/suggestions: In Figure 3, please change the label "2 cycle" and "3 cycle" to "2 cycles" and "3 cycles", respectively.

Remove some "the", such as in Line 151.

Line 149, please comment on the rationale of the 1 g/L concentrations used in this experiment.

Table 1 can be reported by both confirmation of sulfate ions origin (uptake vs H2SO4) and uptake coefficient (before vs after cloud processing).

Line 381: Please replace "were" by "was".

Line 376-377: This sentence contains partly the same data with the above sentences.

Line 539: Please replace "during" by "by".

Line 550-552: Is this a new result? or already known from other studies (then provide proper references)?

In the References, Please exchange "Global Biogeochem, Cycles" by "Global Biogeochem. Cycles".

---

## Referee Comment (RC2) · Anonymous Referee #2 · 1 Aug 2019

General comments

The authors investigated heterogeneous uptake of sulfur dioxide on iron-containing mineral particles. They found enhanced heterogeneous uptake of sulfur dioxide on the mineral particles through modification of iron speciation. The results shown in this paper are very interesting. This study may provide additional pathway to promote the formation of sulfate in iron-containing aerosols. The manuscript is worthy of publication for ACP after some improvements to the readability.

[Figure]

Specific comments

p.4, l.90: The reference of Ito et al. (2019) should be moved to previous sentence after Luo et al. (2008).

p.9, l.205: The formula of geometric area should be corrected.

Table 1: The BET specific surface area is shown in Table S1. The unit and values of the total surface area should be corrected.

p.16, l.373 and Figure 3: Is the trend for ATD statistically significant? Please show the statistics.

Please discuss the results in subsection 3.2 with those in subsection 3.6 as is described in p.21, l.501. Subsection 3.2 may be moved at the end of section 3.

Subsection 3.4 may be combined with subsection 3.1.

p.18, l.430: This has been already suggested by previous studies. Please cite references and rephrase the sentence. Subsection 3.5 may be moved to supplementary materials or before subsection 3.2.

p.20, l.490: This has been already suggested by previous studies. Please cite references and rephrase the sentence. Subsection 3.6 may be moved to supplementary materials or before subsection 3.2.

p.22, l.531: How did you know the particles were coated by reactive Fe? Please specify the relationship between the higher hygroscopicity and reactive Fe coating. Please show this evidence or rephrase the sentence.
* * *

---

## Author Comment (AC1) · 7 Aug 2019

The authors focused on the heterogeneous transformation of SO2 on mineral dust during cloud processing by the simulated experiment. They characterized Fe morphology using the combined methods including TEM, Mossbauer, and CBD extraction method. The iron mobilized from mineral particle was also measured. Generally, heterogeneous transformation on the surface of mineral particle play a vital role on second particle formation. The result shown in this manuscript shed a light on heterogeneous chemistry, and it is helpful to further understand the fog and haze formation in China.

Thus, this manuscript made some new contribution to atmosphere chemistry. The manuscript was well written and organized. The topic is of interest to the journal's readers. I thus recommended that this manuscript could be published on ACP. However, the manuscript also suffered from some flaws, and I listed as follows.

Thank the reviewer's chariness and suggestion. We appreciate the positive comments and suggestion about the manuscript. We agree with the reviewer's comments, and have updated the manuscript on the basis of these suggestions.

General questions/comments/suggestions: 1 The four examined clays were purchased from the clay mineral depository. The author should discuss why they were selected, particularly why they are representative of clays in aerosols. Response: Clay is a more important component of mineral dusts. It was well documented that a long range transport would result in a decrease of quartz relative to the clay fraction because of the more rapid removal of quartz, which has the relatively larger mass median diameter of quartz (Mahowald et al., 2005). Dust deposition modeling showed that the clay fraction becomes more and more dominant on the downwind of the sources from Asia. In the African dust collected over Atlantic Ocean and Mediterranean, the clay group represented between 48 and 82% of the total aerosol mass, illite and kaolinite were the dominant minerals, smectite and chlorite were detected but in low concentrations (Journet et al., 2008). To explore the linkage between iron speciation in the clay minerals and properties of iron solubility, the typical Fe-containing clays, including the Fe-rich nontronite, illite and smectite were selected. Arizona test dust (ATD) is a commercially available material that has also been widely studied as surrogate in the field of atmospheric chemistry and Fe chemistry. The sentence has been added in line 55-58. "A long range transport would result in a decrease of quartz relative to the clay fraction because of the more rapid removal of quartz, hence clay is an important component of mineral dusts (Mahowald et al., 2005; Journet et al., 2008)." The sentence in line 128 has been rewritten as "In this study, we employed four typical Fe-containing mineral samples as surrogates to perform simulated CP experiments."

2 Line 159: "TEM observation". The methods of TEM sample preparation will change the aggregation state of such aggregates. Some researchers have extensively worked on this and gave detailed information how they did it. Nothing is reported here on sample preparation. Therefore, please supply the detail about TEM sample. Response: Thanks for the suggestion. We describe the TEM method in detail in line 214-221. "Suspensions (0.2 g L-1) of each particle were prepared in methanol and sonicated for at least 1 h. A drop of this suspension was then applied to a carbon-coated Cu TEM grid (400 mesh; EMS) and allowed to air-dry. The operation was conducted in bright field mode at 120 kV. The Fe content of the typical individual mineral particle were calculated from the values of 50 typical particles. To obviously observe the morphological changes, high-resolution TEM (HRTEM) images were also collected to observe nanoscale structural features, e.g., surface roughness and lattice fringes."

3 The FeA and FeD content of Arizona Test Dust was "0.067$\pm$ 0.005% for FeA and 0.41$\pm$ 0.04% for FeD" in Shi et al. (2011), which was not completely in accordance with the values in Figure 4. The author should try to explain the discrepancy of FeA and FeD content. Response: In the published paper of Shi et al., the FeA content (0.067$\pm$ 0.005%) and FeD content (0.41$\pm$ 0.04%) were the proportion of FeA and FeD per mass of dust, respectively. The total Fe content (FeT) of Arizona Test Dust (ATD) was 1.48%. The FeA/FeT and FeD/FeT was calculated to be 4.53$\pm$ 0.34% and 27.70$\pm$ 2.70%, respectively, which were comparable to the values in this study, 3.8 $\pm$ 0.3% and 35.5 $\pm$ 3.7%.

Further specific comments/suggestions: In Figure 3, please change the label "2 cycle" and "3 cycle" to "2 cycles" and "3 cycles", respectively. Response: We've changed the label "2 cycle" and "3 cycle" to "2 cycles" and "3 cycles" in Figure 3.

Remove some "the", such as in Line 151. Response: We've removed "the" in line 155.

Line 149, please comment on the rationale of the 1 g/L concentrations used in this experiment. Response: The mineral particle samples are available in large amounts (g-

[Figure]

kg) compared to atmospheric dust ($\mu$g-mg). These samples are clearly the precursor of atmospheric dusts prior to uplifting and thus have not been subject to changes that can happen to mineral dust in the atmosphere, allowing them to be used to investigate how these properties are modified during simulated atmospheric processing. Previous studies examined the iron dissolution over the range of dust loadings from 0.05 to 5 g/L in solutions. The solution with 1 g/L is usually representative for a cloud water solution.

Table 1 can be reported by both confirmation of sulfate ions origin (uptake vs H2SO4) and uptake coefficient (before vs after cloud processing). Response: The IC measurements on sulfate ions released from mineral surface after CP were performed as blank experiments. The sulfate formation rates and uptake coefficients of SO2 on particle samples after CP in Table 1 were recalculated by deduction of the blank value.

Line 381: Please replace "were" by "was". Response: The "were" has been replaced by "was" in line 412.

Line 376-377: This sentence contains partly the same data with the above sentences. Response: Thanks for the reviewer's suggestion. This sentence has been deleted in the revised manuscript.

Line 539: Please replace "during" by "by". Response: The "during" has been replaced by "by" in line 543.

Line 550-552: Is this a new result? or already known from other studies (then provide proper references)? Response: This is a new result of our study.

In the References, Please exchange "Global Biogeochem, Cycles" by "Global Biogeochem. Cycles". Response: We've replaced "Global Biogeochem, Cycles" by "Global Biogeochem. Cycles".

Please also note the supplement to this comment:
https://www.atmos-chem-phys-discuss.net/acp-2019-435/acp-2019-435-AC1-

supplement.pdf

---

## Author Comment (AC2) · 7 Aug 2019

General comments The authors investigated heterogeneous uptake of sulfur dioxide on iron-containing mineral particles. They found enhanced heterogeneous uptake of sulfur dioxide on the mineral particles through modification of iron speciation. The results shown in this paper are very interesting. This study may provide additional pathway to promote the formation of sulfate in iron-containing aerosols. The manuscript is worthy of publication for ACP after some improvements to the readability.

[Figure]

We appreciate the positive comments and suggestion about the manuscript. We agree with the reviewer's comments, and have updated the manuscript on the basis of these suggestion.

Specific comments p.4, l.90: The reference of Ito et al. (2019) should be moved to previous sentence after Luo et al. (2008). Response: The reference of Ito et al. (2019) has been moved to previous sentence after Luo et al. (2008) in line 91.

p.9, l.205: The formula of geometric area should be corrected. Response: The formula of geometric area has been corrected as Ageo = mass $\times$ Sgeo in line 209.

Table 1: The BET specific surface area is shown in Table S1. The unit and values of the total surface area should be corrected. Response: Thanks for the reviewer's correction. The unit and values of ABET has been corrected in Table 1.

p.16, l.373 and Figure 3: Is the trend for ATD statistically significant? Please show the statistics. Response: Although the simulated cloud processing experiment on each mineral was conducted three times to explore the change of Fe speciation after each pH cycle, the SO2 uptake experiment was carried out only twice. At present, it was a pity that we don't have a trend for ATD statistically significant. Next, we'll do experiments more detailed.

Please discuss the results in subsection 3.2 with those in subsection 3.6 as is described in p.21, l.501. Subsection 3.2 may be moved at the end of section 3. Response: Thanks for the suggestion. Because Figure 1 in subsection 3.2 demonstrated that the characteristic peaks for the active OH sites and the sulfite/sulfate products on the mineral particles after CP were significantly higher than those on the ones before CP, indicating the higher hygroscopicity and more SO2 uptake on the particles after CP, which is the most direct evidence that CP could potentially promote the transformation of SO2 on the particle surfaces. In order to emphasize the results and significance of this article, we tend to put subsection 3.2 at the first of section 3.

Subsection 3.4 may be combined with subsection 3.1. Response: We agree with the reviewer's comment. We've combined subsection 3.4 with subsection 3.1 in line 273-308. Thus, the name of every subsection in section 3 has been correspondingly changed.

p.18, l.430: This has been already suggested by previous studies. Please cite references and rephrase the sentence. Subsection 3.5 may be moved to supplementary materials or before subsection 3.2. Response: We've cited reference (Shi et al., 2009) and rephrased this sentence in line 432-436. "Previous research had indicated that FeA increased as a result of the simulated CP (Shi et al., 2009). Herein, we further proposed that the increased fractions of FeA could be mostly transformed from the "structural-Fe" in the aluminosilicate phase of the particles during CP, which is in good agreement with the TEM observation." Subsection 3.5 is one of the most important content to inspect the Fe speciation before and after CP. We want to keep it in the manuscript.

p.20, l.490: This has been already suggested by previous studies. Please cite references and rephrase the sentence. Subsection 3.6 may be moved to supplementary materials or before subsection 3.2. Response: Herein, we've cited references and rephrased the sentence in line 494-495. "The fast Fe release was due to the redissolution of the Fe-rich precipitates, which was proposed to be reactive Fe phases (Shi et al., 2009; Shi et al., 2015)." Subsection 3.6 "The dissolution-precipitation cycle of the mineral Fe during CP" is also one of the most important content to inspect the Fe speciation before and after CP. We tend to keep it in the manuscript.

p.22, l.531: How did you know the particles were coated by reactive Fe? Please specify the relationship between the higher hygroscopicity and reactive Fe coating. Please show this evidence or rephrase the sentence. Response: The dissolution-precipitation cycle of the mineral Fe was happened on the surface of particles. Additionally, the TEM observation confirmed that the nanosized Fe-rich crystal were attached onto the surface of ATD particles. These results help to confirm that the particles after CP

were coated by reactive Fe. To the best of our knowledge, there was no reports about the direct relationship between the higher hygroscopicity and reactive Fe coating. Previous studies have indicated that the reactive Fe could provide more surface hydroxyl species (OH) to participate in chemical reaction (Fu et al., 2007). In the study, the results of DRIFTS experiment demonstrated that the H2O and OH groups on the surface of mineral particles significantly increased after CP, indicating the higher hygroscopicity of the particles after CP. Therefore, we've rephrased the sentence in line 535-537. "The particle surfaces after CP were coated by these reactive Fe to provide more surface OH species, resulting in enhanced SO2 uptake."

Please also note the supplement to this comment:
https://www.atmos-chem-phys-discuss.net/acp-2019-435/acp-2019-435-AC2-supplement.pdf